# Identification and Distribution of Human-Biting Ticks in Northwestern Spain

**DOI:** 10.3390/insects13050469

**Published:** 2022-05-18

**Authors:** María Carmen Vieira Lista, Moncef Belhassen-García, María Belén Vicente Santiago, Javier Sánchez-Montejo, Carlos Pedroza Pérez, Lía Carolina Monsalve Arteaga, Zaida Herrador, Rufino del Álamo-Sanz, Agustin Benito, Julio David Soto López, Antonio Muro

**Affiliations:** 1Infectious and Tropical Diseases Research Group (e-INTRO), Biomedical Research Institute of Salamanca-Research Centre for Tropical Diseases at the University of Salamanca (IBSAL-CIETUS), Faculty of Pharmacy, University of Salamanca, 37008 Salamanca, Spain; carmelilla@usal.es (M.C.V.L.); belvi25@usal.es (M.B.V.S.); s.montejo@usal.es (J.S.-M.); carlospedroza@usal.es (C.P.P.); jdjuliosoto@gmail.com (J.D.S.L.); 2Infectious Diseases Unit, Department of Internal Medicine, University Hospital of Salamanca, 37008 Salamanca, Spain; 3Internal Medicine Department, Ensemble Hospitalier de la Côte, 1110 Morges, Switzerland; liacma@usal.es; 4National Centre for Tropical Medicine, Health Institute Carlos III (ISCIII), 28029 Madrid, Spain; zherrador@isciii.es (Z.H.); abenito@isciii.es (A.B.); 5Consejería de Sanidad Junta Castilla y León, 47007 Valladolid, Spain; alasanru@jcyl.es

**Keywords:** tick bites, Ixodidae, epidemiology: emerging diseases, Spain

## Abstract

**Simple Summary:**

We conducted a tick surveillance study in northwestern Spain. Nymphs of *Ixodes ricinus* were the most frequently collected. *Rhipicephalus bursa*, *Rhipicephalus sanguineus* sensu lato (s.l.), *Hyalomma marginatum*, *Hy. lusitanicum*, *Dermacentor marginatus*, *D. reticulatus* and *Haemaphysalis punctata* were also found, with adults as the main stage. The number of collected *Hyalomma* spp. and *R. bursa* has been progressively increasing over time. Although bites occurred throughout the year, the highest number of incidents was reported from April to July. The distribution patterns of the tick species were different between the north and the south of the region, which was related to cases detected in humans of the pathogens they carried. Adult men were more likely to be bitten by ticks than women. Ticks were most frequently removed from adults from the lower limbs, while for children, they were mainly attached to the head. Epidemiological surveillance is essential given the increase in tick populations in recent years.

**Abstract:**

Ticks transmit a wide diversity of pathogens to a great variety of hosts, including humans. We conducted a tick surveillance study in northwestern Spain between 2014 and 2019. Ticks were removed from people and identified. Tick numbers, species, development stages, the timeline, seasonal and geographical distribution and epidemiological characteristics of people bitten by ticks were studied. We collected ticks from 8143 people. Nymphs of *I. ricinus* were the most frequently collected. *Rhipicephalus bursa*, *R. sanguineus* s.l., *Hy. marginatum*, *Hy. lusitanicum*, *D. marginatus*, *D. reticulatus* and *H. punctata* were also found, with adults as the main stage. The number of collected *Hyalomma* spp. and *R. bursa* has been progressively increasing over time. Although bites occurred throughout the year, the highest number of incidents was reported from April to July. The distribution patterns of the tick species were different between the north and the south of the region, which was related to cases detected in humans of the pathogens they carried. Adult men were more likely to be bitten by ticks than women. Ticks were most frequently removed from adults from the lower limbs, while for children, they were mainly attached to the head. Epidemiological surveillance is essential given the increase in tick populations in recent years, mainly of species potentially carrying pathogens causing emerging diseases in Spain, such as Crimean–Congo hemorrhagic fever (CCFH).

## 1. Introduction

Ticks are hematophagous parasites distributed worldwide, and they are of great importance from an epidemiological and clinical point of view. They can transmit a wide variety of pathogens, such as viruses, bacteria and protozoa [1]. Furthermore, they have a great impact in the veterinary field due to economic losses caused by morbidity and mortality, affecting 80% of the world’s cattle population [2], with the total cost of tick-borne diseases (TBDs) estimated to be between USD 14 and 19 billion globally per year [3]. Tick-borne diseases are a growing public health concern, and their incidence is clearly rising worldwide—just look at the increase in the USA of Lyme and human ehrlichiosis and of tick-borne encephalitis (TBE) and hemorrhagic fever cases in Europe and Asia [4]—due to several interacting factors [5,6]. The ecological characteristics of these vectors affect their epidemiology, with their activity cycles closely related to environmental factors such as temperature and relative humidity that are fundamental for their survival. Climate change and weather variability are just two of the many factors that determine tick population abundance, but there are also other variables of importance, such as composition, host community abundance and landscape features [7,8,9].

Ticks from 896 species are known around the world, and the most prevalent Ixodidae family comprises 702 species in 14 genera. Moreover, ixodid ticks are the main vectors of zoonotic pathogens in Europe [10], where 5 genera (*Ixodes*, *Dermacentor*, *Haemaphysalis*, *Rhipicephalus* and *Hyalomma*) and 54 species have been found, with *I. ricinus* being the most widely distributed tick [9,11,12]. Lyme borreliosis is the most prevalent tick-borne disease in Europe, and it is caused by *Borrelia burgdorferi* sensu lato (s.l.) complex [13]. In the Iberian Peninsula, there are five genera of ixodid ticks that bite humans: *Ixodes*, *Dermacentor*, *Rhipicephalus, Haemaphysalis* and *Hyalomma*, which are potential transmitters of *B. burgdorferi* sensu lato, several genospecies of *Rickettsia*, *Anaplasma phagocytophilum* and the Crimean–Congo virus [14].

The association between global warming and the emergence of TBDs is well documented in Europe and Eurasia [15]. The geographical expansion of tick species is directly related to the emergence of new infectious diseases, which makes it more necessary than ever to understand the dynamics and distribution of ticks. Although there is much previous information about the distribution of ticks in several areas of Spain, the data come mainly from the study of captured ticks in the environment, either from vegetation or animals (wild and domestic) [9,16,17,18,19,20]. The region of Castilla y León is characterized by a great diversity of tick species, as we find both those typical of humid climates and those of Mediterranean climates. There is a very exhaustive study of ticks removed from people in Castilla y León carried out by our group [14] in the period corresponding to 1997–2002. Data on the species collected from humans in this study showed that in the north, *I. ricinus* was the dominant species, followed by both species of *Dermacentor* and *H. punctata.* In the south, in addition to the species mentioned above (except *D. reticulatus*) we found *R. bursa*, *Hy. marginatum* and *R.*
*sanguineus*, which were very abundant in zones rich in livestock and dogs. Although some species are active all year round, most of them are most active in spring–early summer (April–July) and autumn. A long time has passed since then, and given the emergence of new species of ticks and the introduction of new TBDs, such as Crimean–Congo hemorrhagic fever (CCHF) and DEBONEL (*Dermacentor*-borne necrosis erythema and lymphadenopathy), it seems essential to re-examine the current situation in this area of Spain. Thus, the purpose of this work is to update data about species of ticks removed from people and their spatial and temporal patterns. This will allow identifying the risk areas, activity peaks and dynamics of these vectors.

## 2. Materials and Methods

### 2.1. Study Site/Site Selection

This study was performed in Castilla y León (41°23′0″ N, 4°27′0″ W), an area located in northwestern Spain. It covers a surface of 94,224 km^2^ and is one of the most extensive regions of the European Union. Although there is a marked continental climate in most of the territory, characterized by cold winters and hot summers with short periods of spring and autumn, regional variations in both temperatures and rainfall allow us to distinguish different climatic domains in the region: continentalized Mediterranean in the center, with semi-arid enclaves in some areas, mountain Mediterranean in mountainous areas in the northeast, east and south and Atlantic in the north (Figure 1). This great geomorphological and bioclimatic variety, together with its vast extension, gives rise to a great range of climatological conditions and ecosystems which undoubtedly affects the geographical distribution of the different tick species. Castilla y León is rich in forests and ample wooded areas, and the privileged geographical situation of the northern subplateau makes it a region of special interest as an area of passage, breeding and wintering of birds from Central and Northern Europe and the African continent.

### 2.2. Tick Collection and Identification

During 2014–2019, ticks were collected from people who went to primary healthcare centers and hospital emergency services in Castilla y León for their removal through a program of the Junta de Castilla y León for the prevention and control of tick-borne anthropozoonoses. The ticks were removed from the hosts with tweezers and sent to our laboratory (Laboratory of the Faculty of Pharmacy at the University of Salamanca) at room temperature in a container. Each tick was morphologically identified under a binocular lens in terms of life stage and species using taxonomic reference keys [9,21,22,23]. Each tick was identified to the species level, except in the case of *R. sanguineus* (s.l.) ticks, which were identified only to the “group” level as the re-definition of *R. sanguineus* sensu stricto (s.s.) [24] was not available during most of the study. *Rhipicephalus turanicus* specimens were also reported as *R. sanguineus* s.l. Tick species, developmental stages (larva, nymph, adult), sex and feeding degree were recorded, as well as the epidemiological characteristics of the patients (age, sex, geographical location) and anatomic location of the attachment of the tick on the patient. A unique individual identification number was assigned to each tick and its corresponding file.

### 2.3. Geopositioning and Data Analysis

Information about tick bites was obtained from the database of the center of reference of the Biomedical Research Institute of Salamanca-Research Center for Tropical Diseases at the University of Salamanca. To obtain the geographical coordinates of every tick bite, we geocoded the locations of tick collections using “Batch geocoder for journalists” (https://geocode.localfocus.nl, accessed on 28 February 2022), prior to data curation. The latitude and longitude were then projected to the coordinate reference system ETRS89 to map the distribution of the species of interest through the years and the median temperature in the provinces of the autonomous community of Castilla y León, using pretty breaks as a data classification method. All the maps were constructed in QGIS 3.18.3 “Zürich” (Open Source Geospatial Foundation) [25].

Absolute frequencies were used to summarize the number of each species and show visual tendencies across years and seasons. Relative frequencies converted into percentages were used to visualize tendencies between the sex and life stages of the collected ticks, and both frequencies were used to show the epidemiological characteristics of people bitten by ticks. 

In addition, we searched for signals of displacement in the geographic distribution of the collected ticks. For the visual pattern, we used geographic coordinates of the previous geocode step. The comparative figures were constructed using the package *ggplot2* [26], in the statistical environment R [27]. The Kruskal–Wallis rank sum test and pairwise comparisons using the Wilcoxon rank sum test with continuity correction were used for statistical support in R. Chi-square analysis was conducted to evaluate differences in sex specimens. Values of *p* were adjusted using the false discovery rate and Bonferroni methods, considering values less than or equal to 0.05 as statistically significant.

## 3. Results

### 3.1. Tick Numbers, Species, Development Stages and Timeline

A total of 8143 ticks were collected over 6 years, of which 8081 were identified to the species level and life stage (larva, nymph and adult). The remaining 62 were discarded as their morphological identification was impossible due to the structural and conservation state in which they were received. Ixodid ticks belonging to five genera and eight species were found (Table 1); *I. ricinus* was the most frequent species infesting humans (52.29%), followed by *R. bursa* (12.15%), *R. sanguineus* s.l. (10.38%), *Hy. marginatum* (9.08%), *D. marginatus* (8.06%), *Hy. lusitanicum* (3.79%), *D. reticulatus* (2.48%) and *H. punctata* (1.73%). Although this general trend was maintained over the six years of the study, during recent years, there has been a clear increase in *Hy. lusitanicum* and *Rhipicephalus bursa*.

Regarding the temporal distribution (Figure 2), while some species remained stable over time (*D. reticulatus* and *H. punctata*), others underwent changes. *Hyalomma lusitanicum* is a clear example of how a species has increased over time (linear model, R^2^ = 0.9216, F-statistic *p* < 0.05), especially since 2015. Similarly, this occurred with *R. bursa*, with a huge growth spike in 2019 (linear model, R^2^ = 0.6188, F-statistic *p* < 0.05). *Dermacentor marginatus* and *Hy. marginatum* had roughly identical temporal distributions with clear downward trends in 2017 and 2018 before recovering in 2019. *Rhipicephalus sanguineus* and *I. ricinus* both had their maximum peak in 2015. Despite this, both species exhibited a downward trend.

The adult stage (male and female) was the most frequent in all genera (Figure 3) except for *Ixodes*, where nymphs (60.31%) were collected in much larger numbers (Table 1). For *Dermacentor*, *Haemaphysalis* and *Rhipicephalus*, we observed a higher percentage of females compared to males and nymphs (64.24%, 57.85% and 56.72%, respectively) for all the species, while for *Hyalomma* species, the adults recovered were mainly males (68.20%). Larvae were recovered only from *I. ricinus*. This composition of females and males did not change throughout the study. 

### 3.2. Seasonal and Geographical Distribution of Ticks

The highest number of tick bites (78%, *n* = 6303) was reported during spring and summer (April to July) and the lowest in winter (January and early February) (8.90%, *n* = 719). Ixodes ticks were the most frequent and widely distributed species (Figure 4), and although they were detected throughout the year, their highest detection levels were in spring, autumn and early winter for the adult stage and spring and summer for the nymphs (June mainly). Adult stages of the genus *Hyalomma* were removed throughout the year, although very sporadically in winter and autumn, with peak activity in spring and early summer for both species (June and July). *Rhipicephalus* had one activity peak in spring with a decrease in summer (most bites in May and June) and practically disappeared in autumn and winter in the case of *R. bursa*. For *R. sanguineus*, despite the decreases in the number of cases, it was still active during these seasons. Both species of *Dermacentor* were active all year, although they showed pronounced seasonality with two annual peaks of maximum activity in spring and autumn for *D. marginatus* and autumn and winter for *D. reticulatus*, with very little or no presence in summer. Most *D. marginatum* bites were in April and May, with slightly lower peaks in October and March, while most bites by *D. reticulatus* occurred in December and March, with a smaller peak in April. *Haemaphysalis* had a very low occurrence and was limited to spring (May and June).

*Ixodes ricinus* was the species more frequently removed from persons in all seasons. When analyzing the behavior of the remaining species, we observed that *R. sanguineus* and *R. bursa* were the most frequent ticks removed in spring, *Hy. marginatum* and *R. bursa* in summer and *D. marginatus* and *D. reticulatus* in autumn and winter.

The geographical distribution showed variations among the different species of ticks (Figure 5).

Almost all ticks were removed from all provinces of the community except for *D. reticulatus* and *H. punctata*. *Ixodes ricinus* showed a higher prevalence in the northeast, and both *Dermacentor* species in the northern areas, while *Hyalommma* spp. were detected mainly in the south. *Rhipicephalus* spp. showed differences in the distribution of the species, and *R. bursa* was mainly distributed in the south, while *R. sanguineus* was mainly distributed in the south and the northeast. *Haemaphysalis* was the less represented genus, mostly found in the northeast, with little or no presence in the other areas.

Looking at how the distribution of ticks has varied over the years by analyzing the latitude and longitude (Figure 6), we can see there was a tendency to move northwards and westwards when analyzing the data as a group. In the case of the longitude, this was statistically significant both at the group level (χ^2^: 19.422, df = 5, *p* < 0.05) and by year: 2018 compared with 2015 (Z: −3.175819; *p*-adjusted fdr < 0.05), 2019 compared with 2014 (Z: −2.879612; *p*-adjusted fdr < 0.05) and 2019 compared with 2015 (Z: −3.613219; *p*-adjusted fdr < 0.05). In the case of the latitude, this was statistically significant only at the group level (χ^2^: 13.895, df = 5, *p* < 0.05).

### 3.3. Epidemiological Characteristics of People Bitten by Ticks

Data on sex, age and sites of tick bites on people are shown in Table 2. Epidemiological characteristics of the patients were available for 6732 ticks regarding sex, 6460 regarding children/adults and 5736 for age classes. According to the sex of people bitten by ticks, 65.40% were men and 34.59% were women. Ticks were collected from people of all ages: 22.59% of those bitten were children (<14 years), and the remaining 77.39% were adults. The lowest number of bites was found in the 15–35 age group (16.94%). In contrast, the group most frequently bitten by ticks was the 55+ age group (32.98%).

Using the chi-square test (χ^2^ = 339.90, df = 21, *p* < 0.05), we observed statistically significant differences between age groups for each of the species, except in the case of *D. reticulatus*. We removed the greatest number of ticks from the species *I. ricinus*, *R. bursa*, *D. marginatus*, *Hy. marginatum* and *Hy. lusitanicum* from the 55+ age group, while *R. sanguineus* and *H. punctata* were mostly retired from the children’s age group (0–14).

We observed that for all species, the number of ticks removed from men was significantly higher than from women, except in the case of *H. punctata*, in which the number of ticks recovered from women was higher. There were statistically significant differences (χ^2^ = 226.67, df = 7, *p* < 0.05). 

Ticks were most frequently removed from adults from the lower limbs (29.12% of 3561), and children were mainly bitten on the head (38.25% of 1422). *Ixodes ricinus* and *Hyalomma* spp. were removed more frequently from the lower limbs than from other body locations, while *R. sanguineus*., *Dermacentor* spp. and *H. punctata* were mainly fixed on the head. In the case of ticks belonging to the *R. bursa* species, the main attachment site was the thorax.

## 4. Discussion

The emergence of novel tick-borne diseases in recent years has made it essential to understand the distribution of tick populations. Although there are already studies on the distribution of ticks in different areas of Spain [6,14,16,17,18,19,28,29], data on tick species, their distribution and activity have probably undergone changes in recent years. The present study provides epidemiological information on the diversity, relative abundance and seasonal and geographic distribution of human-biting ticks in Castilla y León (northwest Spain) over a six-year span (2014–2019), allowing us to understand tick activity and the periods of highest risk to humans.

*Ixodes ricinus* was the most predominant and widely distributed species, both spatially and temporally, with a clear dominance over the others, followed by *R. bursa*. These results coincide with those observed in other studies carried out by our group seventeen years ago [14], in which *I. ricinus* and *R. bursa* were also the prevalent species. However, data about other ticks have changed over the years. *Hyalomma marginatum* is currently the fourth most frequently removed species from humans, followed very closely by *R. sanguineus* s.l, whereas in a previous study (1996–2002), it was ranked fifth behind *D. marginatus*. Nevertheless, the percentage of recovered ticks belonging to *D. marginatus* was higher in the 1997–2002 period. The most evident case of a change in distribution patterns and abundance was found for *Hy. lusitanicum*, which currently accounts for 4.2% of the ticks removed from humans, compared to 0.85% in the previous study. In this sense, keep in mind that ticks of the genus *Hyalomma* are the main vectors of CCHFV, which is currently considered an emerging or possibly a re-emerging pathogen in Southern Europe. Although the presence of this virus was already reported in Spain in 2010 [30], an increase in human CCHFV cases in Spain and, more specifically, in Castilla y León has been detected, as reported by our group [31,32]. The increase in human cases matched the increase in *Hyalomma* ticks observed in this area. All observed changes in both distribution patterns and frequency of ticks are determined not only by biotic factors such as climate or abiotic factors (vegetation), but also by the accessibility of hosts.

The predominance of *I. ricinus* over the other species is not unique to the study area. The same pattern has been observed in studies carried out in different locations in Spain [14,18,29,33,34,35] and in European countries such as Belgium, Italy, Sweden, the Netherlands, the United Kingdom, Finland, Norway, Romania and Germany. The nymph is the predominant stage of *I. ricinus*, which is also consistent with what was observed in a study previously conducted by our group [14]. Our results compared with those obtained across Europe are lower than those reported in Belgium and Great Britain, higher than those in Italy and similar to those in Sweden [36,37].

Although we cannot conclude that ticks have a preference among human hosts according to their sex, of the 7862 participants included in this study, 65.40% were men and 34.59% were women. This preference for males is very similar to that observed by Fernández Soto in 2003 (62% men and 38% women). The fact that this pattern remains the same as it was almost 20 years ago may indicate that the occupational and behavioral habits in terms of outdoor activities among men and women have not changed. Thus, 22.59% of tick bite victims are in the age group of 14 years old or younger, while the 15–35 group has the lowest number of tick bites at 16.94%, in line with Northern Europe [38] and Belgium [36].

Differences in anatomical sites of attachment were observed for both children and adults. Tick bites occurred most frequently on the lower limbs and thorax in adults and on the head in children. Similar results have been observed in other studies in Western and Northern Europe, with a predominance of bites on the legs in adults and the head and neck in children [13,38,39,40,41]. These apparent preferences, as suggested by some authors, are probably due to the morphological, behavioral and physiological differences between men and women and adults and children. Adult legs and children’s heads are the most accessible places for ticks since they are at the height of the vegetation where ticks search for a host.

Although ticks are removed during all months of the year, the highest number of tick bites are recorded during spring and summer, with a peak of tick activity in June and July, as observed in this same area in previous studies. These data are also in accordance with several studies carried out in Europe [13,38,39,40,42,43,44]. When looking at the temporal distribution of the different genera, we observed that adults of *Ixodes* were detected throughout the year but mostly in spring, autumn and later winter (March), which differed somewhat from previous years in the same study area where adults were biting mainly in autumn and spring but not in winter [14]. These activity patterns are similar to those in several countries of Northern Europe [13,36,42,45]. *Hyalomma* is infrequent in winter, with its peak activity in spring and summer, with June and July being the months with the most reported cases for both species. In recent years, we have seen that *Hy. marginatum* bites are being brought forward to May. The activity pattern of *Rhipicephalus* has been maintained in northwestern Spain over time, with two peaks in spring and summer, although as with *Hyalomma*, in the case of *R. bursa*, there are many reports of bites as early as May. *Dermacentor* shows a particular seasonality, being practically absent in summer for both species, which could lead us to think that it does not feel comfortable with hot temperatures. *Dermacentor marginatus* presents a bimodal activity pattern with peaks in spring (April–May) and autumn, and the peak number of bites always in April. In the case of *D. reticulatus*, bites occur similarly in autumn, winter and spring (from October to March). This differs from the situation observed in Belgium, where the highest peaks of activity occur in spring (March–May) and late summer and autumn (August–November) [36]. *Haemaphysalis* is practically limited to spring. Although previous studies have reported cases in summer, we have not seen them.

While studying the geographical distribution of the ticks, we observed quite stable and different distribution patterns between the north and the south. These distribution patterns were determined by various climatic (temperature and humidity) and ecological factors. *Dermacentor marginatus*, *D. reticulatus* and *H. punctata* were mainly distributed in the north of the study area, although the greatest increase in *D. marginatus* was seen in the southwest. On the other hand, *Hy. marginatum*, *Hy. lusitanicum* and *R. bursa* were found in greater numbers in the south. *Hyalomma marginatum* is expanding its habitat to southwestern areas, and *Hy. lusitanicum* in recent years to arid or semi-arid areas in the west. The wider distribution of *Hyalomma* in the south could explain the increasing occurrence of CCFH virus observed in this area in recent years. Finally, *I. ricinus* and *R. sanguineus* were distributed in both areas. However, the greatest number of specimens was found in the northeast for both, especially *I. ricinus*.

## 5. Conclusions

Our results show that humans from northwestern Spain are mainly bitten by *I. ricinus* nymphs. Ticks bite particularly on the legs of adults and on the head of children. Therefore, although the peak season for tick bites is spring and summer, tick bites are becoming increasingly frequent in autumn and even winter. Seasonal tick patterns have changed in recent years, both in Spain and elsewhere in Europe, where many species of ticks have expanded their distribution. The increased period of activity increases the likelihood of being bitten and therefore of being infected by a tick-borne pathogen. This study is essential for proper epidemiological surveillance. Moreover, knowledge of tick populations and human exposure to tick bites could suggest ways to reduce the risk of tick-borne diseases.

## Figures and Tables

**Figure 1 insects-13-00469-f001:**
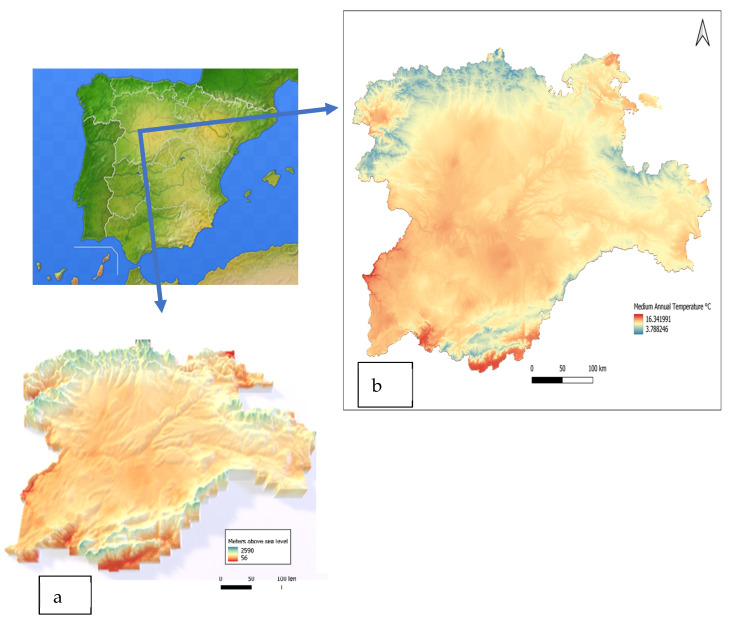
Map of the area of study (Castilla y León, NW Spain). (**a**) Altitude. (**b**) Medium annual temperature. Source: Atlas Agroclimático de Castilla y León—ITACYL-AEMET-2013 (http://atlas.itacyl.es, accessed on 24 April 2022).

**Figure 2 insects-13-00469-f002:**
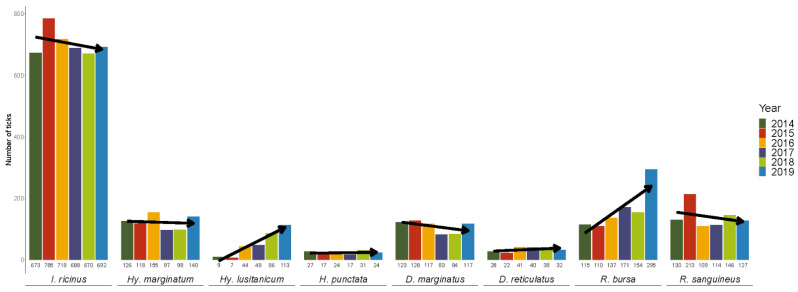
Trends in the number of ticks recovered over time.

**Figure 3 insects-13-00469-f003:**
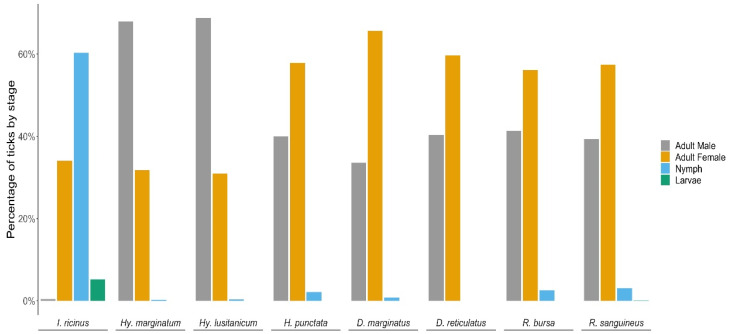
Number of tick species according to developmental stages.

**Figure 4 insects-13-00469-f004:**
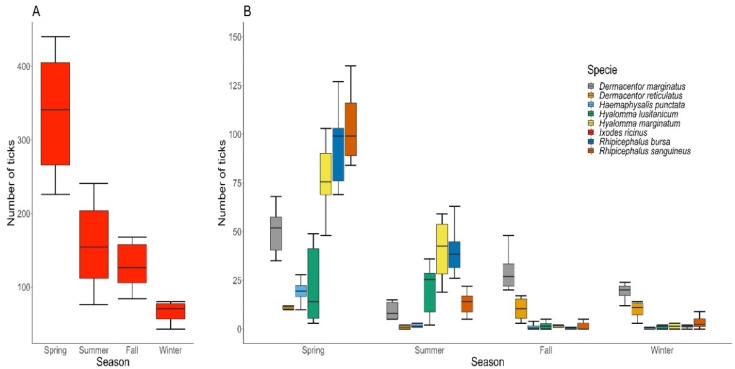
Seasonality of tick bites on humans. (**A**) *I. ricinus*. (**B**) *D. marginatus, D. reticulatus, H. punctata, Hy. Lusitanicum, Hy. Marginatum, R. bursa* and *R. sanguineus*. Species sampled by season and year: winter (December–February), spring (March–May), summer (June–August) and autumn (September–November).

**Figure 5 insects-13-00469-f005:**
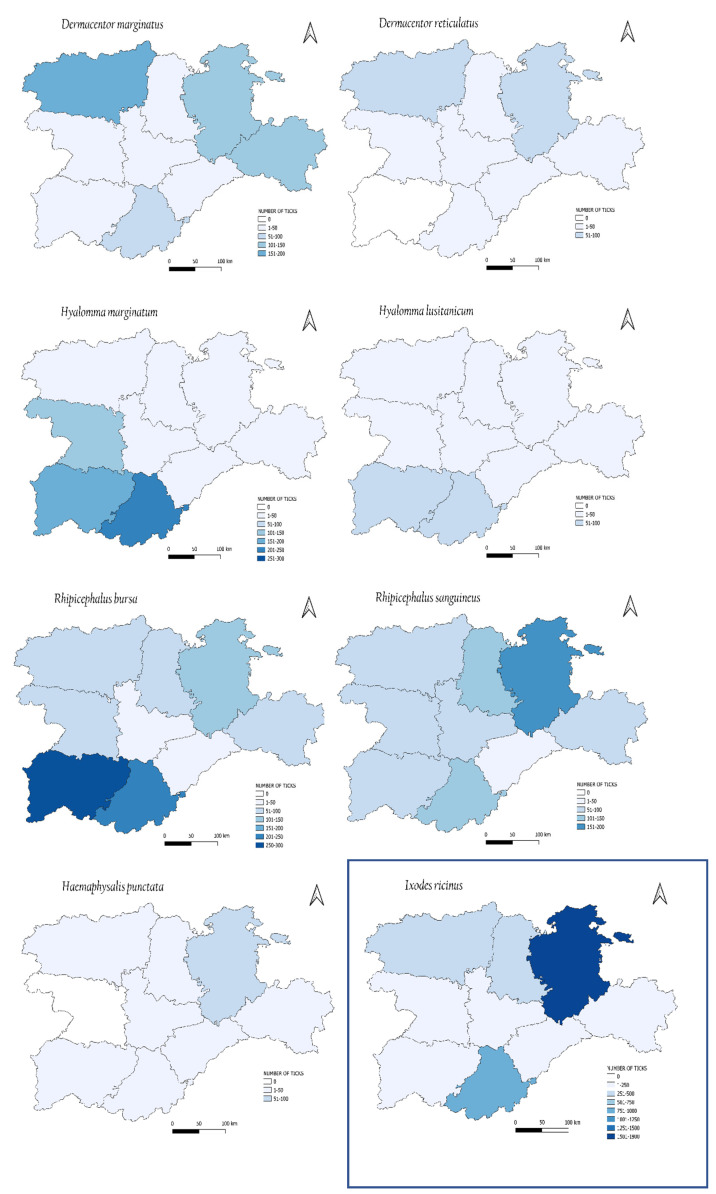
Geographical distribution of ticks. Source: QGIS 3.18.3 “Zürich” (Open Source Geospatial Foundation).

**Figure 6 insects-13-00469-f006:**
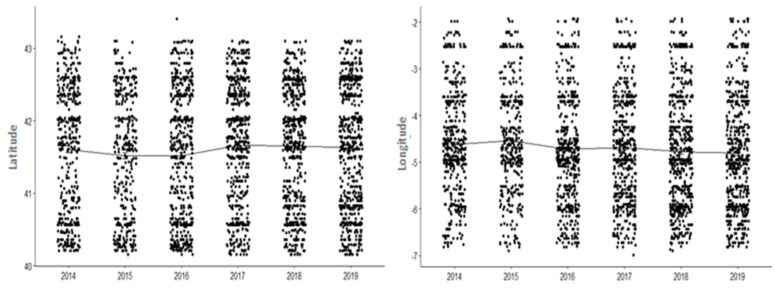
Longitudinal and latitudinal patterns over time.

**Table 1 insects-13-00469-t001:** Numbers of collected ticks, species, life stages and sex.

Tick Species	Numbers of Collected Ticks				
	Larvae	Nymphs	Male	Female	Total
*Ixodes ricinus*	221	2549	18	1438	4226
*Rhipicephalus bursa*		25	406	551	982
*Rhipicephalus sanguineus*	1	26	330	482	839
*Hyalomma marginatum*		2	499	233	734
*Dermacentor marginatus*		5	219	428	652
*Hyalomma lusitanicum*		1	211	95	307
*Dermacentor reticulatus*			81	120	201
*Haemaphysalis punctata*		3	56	81	140
Total	222	2611	1820	3428	8081

**Table 2 insects-13-00469-t002:** Epidemiological characteristics of people bitten by ticks (2014–2019).

Characteristics of Patients	Numbers and Species of Ticks Collected from Patients								
	*I.ricinus*	*R. bursa*	*R. sanguineus*	*D. marginatus*	*D. reticulatus*	*Hy. marginatum*	*Hy. lusitanicum*	*H. punctata*	Total
**Sex: men (%)** **women (%)**	2247 (65.75%) 1170 (34.24%)	633 (73.43%) 229 (26.56%)	411(59.13%) 284 (40.86%)	292 (50.78%) 283 (49.21%)	100 (56.81%) 76 (43.18%)	495 (75.68%) 159 (24.31%)	218 (75.69%) 70 (24.30%)	7 (10.76%) 58 (89.23%)	4403 (65.40%) 2329 (34.59%)
**Total**	**3417**	**862**	**695**	**575**	**176**	**654**	**288**	**65**	**6732**
**Age:**									
**0–14**	673	163	214	126	34	29	13	44	1296 (22.59%)
**15–35**	505	140	108	51	40	81	35	12	972 (16.94%)
**36–55**	840	190	101	134	37	192	61	21	1576 (27.47%)
**55+**	882	228	147	185	26	269	124	31	1892 (32.98%)
**Total**	**2900**	**721**	**570**	**496**	**137**	**571**	**233**	**108**	**5736**
**Adults (%)** **Children (%)**	2478 (75.66%) 797 (24.33%)	615 (75.64%) 198 (24.35%)	408 (60.80%) 263 (39.19%)	403 (73.54%) 145 (26.45%)	124 (75.60%) 40 (24.39%)	580 (94.61%) 33 (5.38%)	240 (93.75%) 16 (6.25%)	70 (58.33%) 50 (41.66%)	4918 (76.13%) 1542 (23.86%)
**Total**	**3275**	**813**	**671**	**548**	**164**	**613**	**256**	**120**	**6460**
**Site of bites:**	**Adults/Children**	**Adults/Children**	**Adults/Children**	**Adults/Children**	**Adults/Children**	**Adults/Children**	**Adults/Children**	**Adults/Children**	**Adults/Children**
**Head**	42/132	54/94	74/151	182/102	28/19	17/9	11/5	16/32	424 (43.80%)/544 (56.19%)
**Neck**	37/59	43/19	46/33	26/8	13/3	29/9	15/5	6/7	215 (60.05%)/143 (39.94%)
**Thorax**	458/159	120/27	60/22	20/7	9/6	100/19	50/9	8/2	825 (76.67%)/251 (23.32)
**Upper limbs**	284/76	41/12	32/14	20/4	12/8	36/8	11/2	6/1	442 (77.95%)/125 (22.04%)
**Lower limbs**	681/188	98/9	39/9	19/5	17/8	125/19	52/8	6/1	1037 (84.65%)/188 (15.34)
**Back**	149/47	46/15	31/11	14/4	8/3	44/8	21/2	4/1	317 (76.69%)/91 (22.30%)
**Pelvis**	147/44	42/14	16/5	4/3	0/0	69/8	18/5	5/1	301 (79%)/80 (20.99%)

## Data Availability

The data presented in this study are available within the article.

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
