# Peer review of "Identification and Distribution of Human-Biting Ticks in Northwestern Spain"

_insects, 2022, doi:10.3390/insects13050469_

Round 1
Reviewer 1 Report
The authors collected ticks from humans presented at clinics over a six-year period in northwestern Spain. Ticks were identified to species and life stage level and data on locality and date was also collected as well as epidemiological parameters of their human hosts. The data were analyzed and presented. A variety of ixodid ticks were recovered with the majority Ixodes ricinus nymphs. For the other tick species, adults were recovered and included Dermacentor marginatus, Dermacentor reticulatus, Haemaphysalis punctatus, Hyalomma lusitanicum, Hyalomma marginatum, Rhipicephalus bursa, and Rhipicephalus sanguineus senso lato. Trends over the time period show decreases and increases depending on species, with important observations on the increase in H. lusitanicum and R. bursa over time. Differences in the recovery of tick sexes and life stages were observed as well as in seasonality with the highest numbers for most species in Autumn, Summer, and Spring. This was related to the life cycle of ticks but also human activity. Differences in geographic localization of species between the north and southern regions of the area of interest were also noted and related to observations on increases in Crimean Congo Hemorrhagic fever in the south. Statistical support for an overall geographic displacement to the north was also inferred. With regard to human hosts, differences in age bitten by ticks were observed with children and 55+ being bitten more. This was also correlated with body sites from which ticks were recovered, with ticks found on the heads of children and legs of adults. This was related to the high of questing of ticks. Males were bitten more than females and this was related to possible occupational risk. The study is of interest since it documents the frequency of exposure to ticks in this region and the associated risk for zoonotic diseases transmitted by ticks. Overall, the study is well written and the conclusions are supported by the data. Some minor issues can be addressed.
Line 32: Ticks transmit …
Line 67: Moreover, ixodid ticks …
Line 101: … undoubtedly, affects the geographic …
Line 130: … prior to data curation.
Line 140: … we searched …
Line 141: … of the collected ticks.
Line 156: Species name in italics.
Line 161, 163, 165, 315, 325, 328: Use full genus name if the sentence starts with the species name.
Line 169: Trends in the number of ticks recovered over time.
Line 185: While the study deals with ticks collected from humans it seems strange that the seasonality is not correlated with the natural life cycle of ticks collected off-host. There must certainly be a correlation between tick abundance, life stages, seasonality, and exposure to humans? Perhaps the authors can show the correlation with ticks collected off-host, or on domestic animals? I am sure that the same trends will be observed as seen for humans. If the authors address this, the message is not clear.
Line 236: Not sure what azX indicates.
Line 331: Species name not capitalized.
References: Genus and species names should be italicized.
Author Response
The authors collected ticks from humans presented at clinics over a six-year period in northwestern Spain. Ticks were identified to species and life stage level and data on locality and date was also collected as well as epidemiological parameters of their human hosts. The data were analyzed and presented. A variety of ixodid ticks were recovered with the majority Ixodes ricinus nymphs. For the other tick species, adults were recovered and included Dermacentor marginatus, Dermacentor reticulatus, Haemaphysalis punctatus, Hyalomma lusitanicum, Hyalomma marginatum, Rhipicephalus bursa, and Rhipicephalus sanguineus senso lato. Trends over the time period show decreases and increases depending on species, with important observations on the increase in H. lusitanicum and R. bursa over time. Differences in the recovery of tick sexes and life stages were observed as well as in seasonality with the highest numbers for most species in Autumn, Summer, and Spring. This was related to the life cycle of ticks but also human activity. Differences in geographic localization of species between the north and southern regions of the area of interest were also noted and related to observations on increases in Crimean Congo Hemorrhagic fever in the south. Statistical support for an overall geographic displacement to the north was also inferred. With regard to human hosts, differences in age bitten by ticks were observed with children and 55+ being bitten more. This was also correlated with body sites from which ticks were recovered, with ticks found on the heads of children and legs of adults. This was related to the high of questing of ticks. Males were bitten more than females and this was related to possible occupational risk. The study is of interest since it documents the frequency of exposure to ticks in this region and the associated risk for zoonotic diseases transmitted by ticks. Overall, the study is well written and the conclusions are supported by the data. Some minor issues can be addressed.
First of all I would like to thank you for your words and comments which will certainly improve the article.
CORRECTED:
Line 32: Ticks transmit
Line 67: Moreover, ixodid ticks …
Line 101: … undoubtedly, affects the geographic …
Line 130: … prior to data curation.
Line 140: … we searched …
Line 141: … of the collected ticks.
Line 156: Species name in italics.
Line 161, 163, 165, 315, 325, 328: Use full genus name if the sentence starts with the species name.
Line 169: Trends in the number of ticks recovered over time
Line 331: Species name not capitalized
Line 185: While the study deals with ticks collected from humans it seems strange that the seasonality is not correlated with the natural life cycle of ticks collected off-host. There must certainly be a correlation between tick abundance, life stages, seasonality, and exposure to humans? Perhaps the authors can show the correlation with ticks collected off-host, or on domestic animals? I am sure that the same trends will be observed as seen for humans. If the authors address this, the message is not clear.
Despite the great variety of ticks existing in Castilla y León, there are only available studies in some provinces on ticks on domestic and wild animals. A lot of research and review work would have to be done. I think this is a very good idea to take into account for future works.
Line 236: Not sure what azX indicates.
This is a mistake. The table leyend should be there.
References: Genus and species names should be italicized.

Reviewer 2 Report
Dear Authors,
the tick data you collected is impressive. Unfortunately, I have to recommend the rejection of your manuscript in its present form. There are too many issues (mistakes and contradictions). The manuscript needs rewriting, more data, tables, and better analyses (your results are not supported by statistical tests). Below, please find my comments (up to discussion). I have to admit that there was no point in reviewing the discussion with the current form of results. The manuscript requires still much work but after improvements, I believe it can be of great value. I wish you good luck.
Figures: please do not describe the results in the titles.
Line 22-23 and 37-38: When using the genus name for the second time please use the abbreviation “R. sanguineus, …, Hy. lusitanicum, …, D. rerticulatus”
Line 23 and 37: “(s.l.)” a dot after “l” is missing
Line 34: “..from people and identified”
Line 34-35 and elsewhere: What do the authors mean by “evolution over time”? The authors did not perform any phylogenetic analyses.
Lines 24-25 and 39-40: Do the authors mean “The numbers of collected … have been progressively increasing over time”?
Lines 47, 74, 84, and elsewhere: Please unify the writing of “Crimean–Congo”. No need to use capital letters for “hemorrhagic fever”.
The introduction needs improvement. It is too generic and the manuscript would profit from a more detailed introduction, e.g. data on the ecology of ticks (what is the main host range for species you found?), distribution (are all species endemic to this area?), and activity of tick species found in your study. The activity pattern would explain the infestation pattern of patients.
Line 54: Maybe replace the second “due to” with “caused by”?
Line 58: Maybe use “rising” instead of “increasing” to avoid repetition?
Line 58 and 59: Please use consistent spacing before/after “-“
Line 59: Please introduce the full name of TBE
Line 62: “… humidity that are…”
Lines 71-73: “On the Iberian….” - this should be a part of the results. Please consider rewriting it or remove from this section.
Line 75: “TBDs”
Line 80: “… either from vegetation or animals…”
Line 88, 89: “This will allow to…”, “… these vectors”
Line 93: “94,224” please use a comma instead of a dot (dot indicates a decimal). “… is one of the most extensive regions of…” - Övre Norrland in Sweden and Pohjois-Suomi in Finland are larger
Line 101: “which undoubtedly affect…”
Fig. 1. Please consider uploading maps in a better resolution. Additionally, fig. 1.a is missing a legend and the legend of 1.b is hard to read, 1.b is also stretched and therefore deformed. Please provide the source of the maps (QGIS?). No need to use uppercase for “medium annual temperature”.
Line 117: “s.l.”
Line 118-19: “sensu stricto (s.s.)”
Line 129: “geocoded”
Line 134: “…done in QGIS 3.18.3 “Zürich” (Open Source Geospatial Foundation) [25]”
Line 135: “of each species”
Line 136: “tendencies across years and seasons”
Line 137: “were used to visualize…”, “between sex and life stages of collected ticks, and both…”
Line 140: “searched”
Line 142: “ggplot2” in italics
Line 143: “pairwise” no need of uppercase
Line 150-151: “which 7345 were identified to the species level and life stage.”
Results: Please provide a table with numbers of collected ticks species, life stages and sex. In the text, the authors mention % of collected ticks but there are no numbers of collected ticks (XY%; n=…) (lines 155-7).
Line 155: “…infesting humans…”
Lines 160-166: This section is missing data (numbers and statistical tests). Please introduce the respective values and refer to the figure in the first place you mention its data.
Line 162: “a species”
Fig.2. I suggest using different colors than shades of one color for better visualization. Please change “evolution over time” – this is the wrong phrase to use for what you are presenting. More appropriate would be e.g. “Number of collected tick species over time”. Please delete the species names from the title of the figure. You can use Ha./Hy. to distinguish between two genera
Lines 173-179: “… in all genera except for Ixodes…”. Please provide the respective numbers of tick life stages and sex (I suggest adding a table and then referring to it) as well as the test value and p-value. You could also mention that larvae were recovered only from I. ricinus. “These sex differences were statistically significant (P < 0.05)” – do you mean only females and males or were there differences between females, males, nymphs and larvae? Please clarify.
Lines 177-9: “When we analysed the interspecies percentage of females and males, we observed no intraspecies variability (Figure 3)” – this does not make sense, please help me understand: when looking at differences between the species, you did not find differences within a species? Do you mean that the composition of females and males did not change over time for each species? And, this statement is missing statistics.
Figure 3. “Number of collected tick species according to developmental stages.” Please delete the rest of the title.
Line 186: “The highest number of tick bites (69% of the total)” – please add the number
Lines 186-207: This section needs rewriting. Consider adding information about which tick species were active all year, which had the peak in the respective seasons. Were there ticks collected only in one season? The authors mention the season activity for different life stages only for I. ricinus – please delete it or provide such information for the other species.
“The highest number of all tick bites (69%, n=…) was reported during spring and summer (April to July) and the lowest in winter (%, n). Ixodes ticks were the most frequent and widely distributed species throughout the year, while all other species showed marked seasonality” - it is not true as Ixodes also show seasonality: the activity peak is in spring. “Although Ixodes spp. was detected throughout the year, its highest activity was in spring”.
Figure 4. The legend is unreadable. Please use a bigger font size. “All ticks belonging to each genus showed a repetitive pattern over time” – this does not result from this figure, please delete it from the title.
Lines 199-207: “Rhipicephalus had two peaks of activity in spring and summer and then practically disappeared in autumn and winter” – this is one activity peak with a decrease during the year. If you provide % for some species, please do it for all. For Dermacentor reticulatus I would rather say it was active all year with a decrease in summer.
Paragraphs 186-192 and 199-207 should be merged.
Figure 5. Please upload pictures in better quality, they are very hard to read. Please either unify the scale between the maps or introduce different colors for different ranges, right now this is misleading. Please remove the description from the title, and add the source of maps (QGIS?)
Lines 215-217: Were other species not detected in all provinces as you stated for Hyalomma spp.? There is no need for a new paragraph (lines 218-21 should be together with 215-17)
Lines 222-226: “Looking at how the distribution of ticks… analysing latitude and longitude for a group (Figure 6), we can see that there is a tendency…”. “In the case of longitude, this is statistically significant both at the group level (test and p values) and by year (test and p values), especially between 2014 and 2015 225 with 2019 (P< 0.05 pairwise comparison)”. What about the statistical differences for latitude?
Table 1. I believe that due to confusion during the last phase before submission, the authors inserted the wrong format of the table. Please provide the table title, and the correct format of the table (not a screenshot so words are not underlined due to spelling settings of the program). The table is missing a headline explaining the numbers (e.g. “number of ticks collected from patients (n, %), names for the first column (e.g. “characteristics of patients”) and row (e.g. “tick species recovered from patients”). Please correct the spelling of “species”, “I. ricinus”. I recommend adding a row with “total”. The values in columns do not sum up to 100% (e.g. for I. ricinus: male/men 67.5% + female/women 34.24% = 101.74%). Also when summing up numbers (e.g. for I. ricinus), in each category, there is a different total number (e.g. for sex: 3417, age 2900, adult/child: 3275), please check your calculations again. I suggest using fractions to the first decimal place. Please use a dot instead of a comma for decimals. The table doesn’t show that children had most of the ticks bites on the head (also lines 241-2). Please use “men” and “women” when describing the sex of people. “Site” is a wrong category for “adult/child”. Please provide the %-values for all categories, also age and body parts. What does the “/” mean in the body parts category, e.g. head: 42/132 (for I. ric.), lower limbs: 681/188 (for I. ric.), j19/5 (for D. mar.)?
Lines 231-5: The numbers in the paragraph are different than in the table
Lines 238-40: Please provide some statistics for these statements. For I. ricinus, it looks like age groups 36-55 and 55+ are mostly infested with ticks (based on the table). What tick life stages were dominating in each age group? Was there any trend?
Lines 241-244: There is no data shown supporting this, either in the table or text.
Author Response
The tick data you collected is impressive. Unfortunately, I have to recommend the rejection of your manuscript in its present form. There are too many issues (mistakes and contradictions). The manuscript needs rewriting, more data, tables, and better analyses (your results are not supported by statistical tests). Below, please find my comments (up to discussion). I have to admit that there was no point in reviewing the discussion with the current form of results. The manuscript requires still much work but after improvements, I believe it can be of great value. I wish you good luck.
First of all, I would like to thank you for all your comments and corrections, which will undoubtedly help to improve the article considerably.
CORRECTED:
-Figures: please do not describe the results in the titles.
-Line 22-23 and 37-38: When using the genus name for the second time please use the abbreviation “R. sanguineus, …, Hy. lusitanicum, …, D. rerticulatus”
-Line 23 and 37: “(s.l.)” a dot after “l” is missing.
-Line 34: “..from people and identified”.
Lines 47, 74, 84, and elsewhere: Please unify the writing of “Crimean–Congo”. No need to use capital letters for “hemorrhagic fever”.
Line 54: Maybe replace the second “due to” with “caused by”?
Line 58: Maybe use “rising” instead of “increasing” to avoid repetition?
Line 58 and 59: Please use consistent spacing before/after ““
Line 59: Please introduce the full name of TBE
Line 62: “… humidity that are…”
Line 75: “TBDs”
Line 80: “… either from vegetation or animals…”
Line 88, 89: “This will allow to…”, “… these vectors”
Line 101: “which undoubtedly affect…
Fig. 1. Please consider uploading maps in a better resolution. Additionally, fig. 1.a is missing a legend and the legend of 1.b is hard to read, 1.b is also stretched and therefore deformed. Please provide the source of the maps (QGIS?). No need to use uppercase for “medium annual temperature”.
Line 117: “s.l.”
Line 118-19: “sensu stricto (s.s.)”
Line 129: “geocoded”
Line 134: “…done in QGIS 3.18.3 “Zürich” (Open Source Geospatial Foundation) [25]”
Line 135: “of each species”
Line 136: “tendencies across years and seasons”.
Line 137: “were used to visualize…”, “between sex and life stages of collected ticks, and both…”,
Line 140: “searched”
Line 142: “ggplot2” in italics
Line 143: “pairwise” no need of uppercase
Line 150-151: “which 7345 were identified to the species level and life stage.”
Results: Please provide a table with numbers of collected ticks species, life stages and sex. In the text, the authors mention % of collected ticks but there are no numbers of collected ticks (XY%; n=…) (lines 155-7).
Line 155: “…infesting humans…
Lines 160-166: This section is missing data (numbers and statistical tests). Please introduce the respective values and refer to the figure in the first place you mention its data.
Line 162: “a species”
Fig.2. I suggest using different colors than shades of one color for better visualization. Please change “evolution over time” – this is the wrong phrase to use for what you are presenting. More appropriate would be e.g. “Number of collected tick species over time”. Please delete the species names from the title of the figure. You can use Ha./Hy. to distinguish between two genera.
Figure 3. “Number of collected tick species according to developmental stages.” Please delete the rest of the title.
Line 186: “The highest number of tick bites (69% of the total)” – please add the number.
Figure 4. The legend is unreadable. Please use a bigger font size. “All ticks belonging to each genus showed a repetitive pattern over time” – this does not result from this figure, please delete it from the title.
Lines 199-207: “Rhipicephalus had two peaks of activity in spring and summer and then practically disappeared in autumn and winter” – this is one activity peak with a decrease during the year. If you provide % for some species, please do it for all. For Dermacentor reticulatus I would rather say it was active all year with a decrease in summer.
Paragraphs 186-192 and 199-207 should be merged.
Line 93: “94,224” please use a comma instead of a dot (dot indicates a decimal). “… is one of the most extensive regions of…” - Övre Norrland in Sweden and Pohjois-Suomi in Finland are larger.
Lines 231-5: The numbers in the paragraph are different than in the table
-Lines 24-25 and 39-40: Do the authors mean “The numbers of collected … have been progressively increasing over time”?
Yes, of course it refers to ticks removed. It is true that it was expressed in an unclear way.
-Lines 186-207: This section needs rewriting. Consider adding information about which tick species were active all year, which had the peak in the respective seasons. Were there ticks collected only in one season? The authors mention the season activity for different life stages only for I. ricinus – please delete it or provide such information for the other species.
Season activity for diferente life stages is only discussed in the case of I. ricinus because it is the only species for which we have relevant data on stages other than the adult stage.
-Lines 222-226: “Looking at how the distribution of ticks… analysing latitude and longitude for a group (Figure 6), we can see that there is a tendency…”. “In the case of longitude, this is statistically significant both at the group level (test and p values) and by year (test and p values), especially between 2014 and 2015 225 with 2019 (P< 0.05 pairwise comparison)”. What about the statistical differences for latitude?
In the case of latitude this is statistically significant only at the group level (Kruskall- Wallis rank test; P< 0.05 pairwise comparison)
-Line 34-35 and elsewhere: What do the authors mean by “evolution over time”? The authors did not perform any phylogenetic analyses.
We refer to how their distribution and frequency has changed over time. Perhaps it would be convenient to use another term to avoid confusion.
-The introduction needs improvement. It is too generic and the manuscript would profit from a more detailed introduction, e.g. data on the ecology of ticks (what is the main host range for species you found?), distribution (are all species endemic to this area?), and activity of tick species found in your study. The activity pattern would explain the infestation pattern of patients.
Following your suggestions I have introduced some lines about existing data on the distribution and activity of ticks in the area.
-Lines 71-73: “On the Iberian….” - this should be a part of the results. Please consider rewriting it or remove from this section.
This is not a result.
-Lines 173-179: “… in all genera except for Ixodes…”. Please provide the respective numbers of tick life stages and sex (I suggest adding a table and then referring to it) as well as the test value and p-value. You could also mention that larvae were recovered only from I. ricinus. “These sex differences were statistically significant (P < 0.05)” – do you mean only females and males or were there differences between females, males, nymphs and larvae? Please clarify.
I have added a table with the requested data, and I have removed the statistics because I realised that they were not comparable groups as in some we had 4 classes and in others only 2. We are therefore only talking about percentages of the different stages.
-“The highest number of all tick bites (69%, n=…) was reported during spring and summer (April to July) and the lowest in winter (%, n). Ixodes ticks were the most frequent and widely distributed species throughout the year, while all other species showed marked seasonality” - it is not true as Ixodes also show seasonality: the activity peak is in spring. “Although Ixodes spp. was detected throughout the year, its highest activity was in spring”.
You are absolutely right. I have already modified it in the text.
Figure 5. Please upload pictures in better quality, they are very hard to read. Please either unify the scale between the maps or introduce different colors for different ranges, right now this is misleading. Please remove the description from the title, and add the source of maps (QGIS?)
Lines 215-217: Were other species not detected in all provinces as you stated for Hyalomma spp.? There is no need for a new paragraph (lines 218-21 should be together with 215-17)
Modified
Table 1. I believe that due to confusion during the last phase before submission, the authors inserted the wrong format of the table. Please provide the table title, and the correct format of the table (not a screenshot so words are not underlined due to spelling settings of the program). The table is missing a headline explaining the numbers (e.g. “number of ticks collected from patients (n, %), names for the first column (e.g. “characteristics of patients”) and row (e.g. “tick species recovered from patients”). Please correct the spelling of “species”, “I. ricinus”. I recommend adding a row with “total”. The values in columns do not sum up to 100% (e.g. for I. ricinus: male/men 67.5% + female/women 34.24% = 101.74%).
You are absolutely right. It was a mistake. The table was not properly formatted and it was not the final table. I think and I hope the error has been fixed.
- Also when summing up numbers (e.g. for I. ricinus), in each category, there is a different total number (e.g. for sex: 3417, age 2900, adult/child: 3275), please check your calculations again.
Unfortunately we do not have all the data for all patients. In many cases some of the epidemiological characteristics are missing. That is why not all figures coincide and why percentages are made for each section.
- Please use a dot instead of a comma for decimals. The table doesn’t show that children had most of the ticks bites on the head (also lines 241-2).. Please use “men” and “women” when describing the sex of people. “Site” is a wrong category for “adult/child”. Please provide the %-values for all categories, also age and body parts. What does the “/” mean in the body parts category, e.g. head: 42/132 (for I. ric.), lower limbs: 681/188 (for I. ric.), j19/5 (for D. mar.)?
Corrected.
Lines 238-40: Please provide some statistics for these statements. For I. ricinus, it looks like age groups 36-55 and 55+ are mostly infested with ticks (based on the table). What tick life stages were dominating in each age group? Was there any trend?
Modified and completed
Lines 241-244: There is no data shown supporting this, either in the table or text

Round 2
Reviewer 2 Report
Dear Authors,
thank you for your adjustments. The paper reads now much easier. I still spotted some issues (e.g. misspellings or double spacing). Please check your manuscript for those errors. Please find my comments below (regarding e.g. wrong interpretation of statistical tests, missing figure). Please add figures/maps/graphs in better quality, they are very hard to read. Again, I didn’t go through the discussion as the results are still not optimal. I suggest a semi-major revision.
Lines 22-23: Please change to: “Rhipicephalus bursa, R. sanguineus sensu lato (s.l.), Hyalomma marginatum, Hy. lusitanicum, Dermacentor marginatus, D. reticulatus"
Lines 34-35, 333: please replace “evolution”
Lines 37-38: Please change to: “Rhipicephalus bursa, R. sanguineus s.l., Hy. marginatum, Hy. lusitanicum, D. marginatus, D. reticulatus and H. punctata"
Lines 24-25 and 39-40: “The number of collected … has been progressively increasing over time” – please use singular “has” for “number” or plural “have” for “numbers”
Line 75: missing “n” in Crimean-Congo (“Crimea-Congo”)
Lines 71-73: “On the Iberian….” - this should be a part of the results. Please consider rewriting it or remove from this section.
- This is not a result.
- The Authors write “we found” which indicates that these are the results. Maybe consider rephrasing “On the Iberian Peninsula, there are 5 genera of Ixodid ticks that bite humans”?
- Also in lines 68-69, you write “we found 5 genera and 54 species” – in your study, you found 8 species. Do you mean that in Spain, there are 54 tick species from 5 genera (are they the same genera that bite humans?)?
- Please use a consistent numbering style - “5/five”
Line 73-74: “Borrelia burgdorferi s.l., several species of Rickettsia”
Line 82: “The region of Castilla… is characterized by a great diversity of tick species…”
Line 86: “I. ricinus is the dominant species”. It is not endangered as “most at risk” indicates.
Line 97: “ these vectors” not “vector”
Line 110: please use a space after “affects”
Fig. 1.: it still needs a better resolution. The legend in 1.a is missing and the legend of 1.b is very hard to read. Please change “1.a” to “1a” or “1b” to “1.b”.
Line 129: “s.l.” please remove a space
Line 130: please add a space after “(s.s.)”
Line 146: please add a space between “3.18.3” and “Zürich”
Line 148: please add a space between “each” and “species”
Line 150: please remove space before a comma. There is also double spacing in this line
Lines 167-169: please keep to one digit after a dot in decimals. “s.l.” – dot after “l”
Table 1. There is no need to divide life stage and sex into different columns. Please merge cells “life stage” and “sex” into one “Numbers of collected ticks: larvae, nymphs, males, females”. In the last row of the table, there is a missing “total”. I suggest also adding %-values to the table. And to keep either alphabetic order of species or from the most abundant to the least.
Line 175: “D. reticulatus” – a dot is missing
Line 177 and 178: p>0.05 indicates that models are not significant so even though the model explains a lot of variation (high R-square), it is worthless as the p-value is too high. Therefore, the statements in this paragraph are not true (the clear increase over time for mentioned species)
Figure 2 is missing
Lines 191-193: This statement is repeating the one from lines 187-190. Please delete it.
Figure 3 is not cited in the text. It can be removed.
Line 199: 8081 is a total of tick bites, thus 78% of 8081 tick bites is 6303. Please write either (78% of 8081) or (78%, n=6303). The same for the lowest. “The highest number of tick bites (78%, n=6303) was reported during spring and summer (April to July) and the lowest in winter (8.9%, n=719)”. You could also name the winter months for the lowest tick activity as you did for spring and summer.
Line 201: a space is missing
Line 202: “Late summer” not winter
Line 203: “adult stages”
Line 211: Please delete % or provide them for other species and periods.
Lines 211-213: if you provide monthly bite info for Dermacentor, you should do this for other species as well (Hyalomma and Rhipicephalus).
Line 216: “R. sanguines"- please add a space
Line 219: please remove “=”
Figure 4. Please provide a graph of better quality. The legend is unreadable. Please use a bigger font size.
Figure 5. Please upload bigger maps, they are very hard to read.
Line 280: “Almost all tick species were removed…”
Lines 281-286: there are dots and spaces missing. Please write: “Ixodes ricinus showed a higher prevalence in the northeast, both Dermacentor species in the northern areas while Hyalommma spp. were detected mainly in the south. Rhipicephalus spp. Showed differences in the distribution of the species, R. bursa was mainly distributed in the south, while R. sanguineus in the south and the northeast”.
Lines 291-294: In the results do not write the name of the test, just a value of the test (which indicates the test): H, df, and p-value for Kruskal-Wallis, and for Wilcoxon: T, Z, and p.
Figure 6: please change the title
Paragraph 3.3: please state that the data about characteristics of patients were available only for 6795 ticks regarding sex, 6460 regarding children/adults and 5736 for age classes.
Lines 303: the %-values are different in the table and text for children and adults
Table 2. Please provide %-values for all numbers in the table (why is it only for sex and adult/children). Please change the heading to “Numbers (n, %) and species of ticks collected from patients” and add the missing %. Please provide the total number of ticks for which data was available for each section (the one in the headings is misleading as it is different for each section)– the best would be to insert additional rows after each section with (sub)total numbers and %.
Lines 311-319: I am not sure if the Chi-square test is appropriate to calculate this. I believe it should be the t-test for comparing two groups (e.g. men vs. women) and ANOVA for more than 2 groups (e.g. age classes). Please calculate it again and rewrite the paragraph
Line 321-322: “(29.1%, n=1037)” 3561 is the total number of bites on adults not from lower limbs. For children, it should also be changed “38.3%, n=544”.
Author Response
Thank you for your adjustments. The paper reads now much easier. I still spotted some issues (e.g. misspellings or double spacing). Please check your manuscript for those errors. Please find my comments below (regarding e.g. wrong interpretation of statistical tests, missing figure). Please add figures/maps/graphs in better quality, they are very hard to read. Again, I didn’t go through the discussion as the results are still not optimal. I suggest a semi-major revision.
Thank you again for your comments and revisions which are very helpful
Lines 22-23: Please change to: “Rhipicephalus bursa, R. sanguineus sensu lato (s.l.), Hyalomma marginatum, Hy. lusitanicum, Dermacentor marginatus, D. reticulatus"
Lines 34-35, 333: please replace “evolution” E
Lines 37-38: Please change to: “Rhipicephalus bursa, R. sanguineus s.l., Hy. marginatum, Hy. lusitanicum, D. marginatus, D. reticulatus and H. punctata"
Lines 24-25 and 39-40: “The number of collected … has been progressively increasing over time” – please use singular “has” for “number” or plural “have” for “numbers”
Line 75: missing “n” in Crimean-Congo (“Crimea-Congo”)
Lines 71-73: “On the Iberian….” - this should be a part of the results. Please consider rewriting it or remove from this section.
- This is not a result.
- The Authors write “we found” which indicates that these are the results. Maybe consider rephrasing “On the Iberian Peninsula, there are 5 genera of Ixodid ticks that bite humans”?
You are right. We have deleted ¨found
- Also in lines 68-69, you write “we found 5 genera and 54 species” – in your study, you found 8 species. Do you mean that in Spain, there are 54 tick species from 5 genera (are they the same genera that bite humans?) DONE
I meant ticks from Europe. I have changed it to make it clearer.
- Please use a consistent numbering style - “5/five”
Line 73-74: “Borrelia burgdorferi s.l., several species of Rickettsia”
Line 82: “The region of Castilla… is characterized by a great diversity of tick species…”
Line 86: “I. ricinus is the dominant species”. It is not endangered as “most at risk” indicates.
Line 97: “ these vectors” not “vector”
Line 110: please use a space after “affects”
Fig. 1.: it still needs a better resolution. The legend in 1.a is missing and the legend of 1.b is very hard to read. Please change “1.a” to “1a” or “1b” to “1.b”.
Line 129: “s.l.” please remove a space
Line 130: please add a space after “(s.s.)”
Line 146: please add a space between “3.18.3” and “Zürich”
Line 148: please add a space between “each” and “species”
Line 150: please remove space before a comma. There is also double spacing in this line
Lines 167-169: please keep to one digit after a dot in decimals. “s.l.” – dot after “l”.
Table 1. There is no need to divide life stage and sex into different columns. Please merge cells “life stage” and “sex” into one “Numbers of collected ticks: larvae, nymphs, males, females”. In the last row of the table, there is a missing “total”. I suggest also adding %-values to the table. And to keep either alphabetic order of species or from the most abundant to the least.
We felt it was overloading the table and these data are reflected in the text.
Line 175: “D. reticulatus” – a dot is missing
Line 177 and 178: p>0.05 indicates that models are not significant so even though the model explains a lot of variation (high R-square), it is worthless as the p-value is too high. Therefore, the statements in this paragraph are not true (the clear increase over time for mentioned species)
Sorry with confusion, it was a mistake in the sign
Figure 2 is missing.
Lines 191-193: This statement is repeating the one from lines 187-190. Please delete it.
Figure 3 is not cited in the text. It can be removed.
Thank you very much for the observation. We have already cited, and we think this is a good figura that at a glance makes clear yhe composition of the different stages for each specie so we would prefer to keep it.
Line 199: 8081 is a total of tick bites, thus 78% of 8081 tick bites is 6303. Please write either (78% of 8081) or (78%, n=6303). The same for the lowest. “The highest number of tick bites (78%, n=6303) was reported during spring and summer (April to July) and the lowest in winter (8.9%, n=719)”. You could also name the winter months for the lowest tick activity as you did for spring and summer.
Line 201: a space is missing
Line 202: “Late summer” not Winter
Really is early summer
Line 203: “adult stages”
Line 211: Please delete % or provide them for other species and periods.
Lines 211-213: if you provide monthly bite info for Dermacentor, you should do this for other species as well (Hyalomma and Rhipicephalus).
Line 216: “R. sanguines"- please add a space
Line 219: please remove “=”
Figure 4. Please provide a graph of better quality. The legend is unreadable. Please use a bigger font size.
Figure 5. Please upload bigger maps, they are very hard to read.
Line 280: “Almost all tick species were removed…”
Lines 281-286: there are dots and spaces missing. Please write: “Ixodes ricinus showed a higher prevalence in the northeast, both Dermacentor species in the northern areas while Hyalommma spp. were detected mainly in the south. Rhipicephalus spp. Showed differences in the distribution of the species, R. bursa was mainly distributed in the south, while R. sanguineus in the south and the northeast”.
Lines 291-294: In the results do not write the name of the test, just a value of the test (which indicates the test): H, df, and p-value for Kruskal-Wallis, and for Wilcoxon: T, Z, and p.
Figure 6: please change the title
Paragraph 3.3: please state that the data about characteristics of patients were available only for 6795 ticks regarding sex, 6460 regarding children/adults and 5736 for age clases
Regarding sex there are 6732 (when recalculating all totals we detected a mistake)
Lines 303: the %-values are different in the table and text for children and adultsh.
You are right. Now is correct. Thank you
Table 2. Please provide %-values for all numbers in the table (why is it only for sex and adult/children). Please change the heading to “Numbers (n, %) and species of ticks collected from patients” and add the missing %. Please provide the total number of ticks for which data was available for each section (the one in the headings is misleading as it is different for each section)– the best would be to insert additional rows after each section with (sub)total numbers and %.
I have introduced some of the suggested changes, but not all. I do not indicate the percentages of all the values because apart from the fact that the table is very overloaded with data, these percentages can be made both column and row and in some cases (part of the body) there are two values in each cell and it would be necessary to add both individual % and totals per row and column. It seems to us that this is too much data and we think that with the table and the text it is quite clear.
Lines 311-319: I am not sure if the Chi-square test is appropriate to calculate this. I believe it should be the t-test for comparing two groups (e.g. men vs. women) and ANOVA for more than 2 groups (e.g. age classes). Please calculate it again and rewrite the paragraph.
We agree that the Chi-square test is not a particularly robust test, but it is the only statistical tool we have been able to use to deal with the data as we have it. The suggested tests have been impossible to apply due to the type of data. It is not possible to perform an analysis of variance as we did not generate sample replicates to obtain a mean and variance. Our experimental design is set up in such a way that we work with absolute values. It is for this reason that we initially did not include any statistical tests.
Line 321-322: “(29.1%, n=1037)” 3561 is the total number of bites on adults not from lower limbs. For children, it should also be changed “38.3%, n=544”.

Round 3
Reviewer 2 Report
Dear Authors,
there are still some last minor issues that I spotted. I do not need to receive the manuscript again after you adjust the paper.
In figures 2, 3 and table 2 and the discussion please use “Hy.” instead of “H.” for Hyalomma species (Lines 37, 177, 215, 305, 350, 368, 370 and elsewhere)
Line 74: please remove a dot before a comma “sensu lato, several…”
Table 1 is missing
In table 2, number 52 is unnecessarily bolded (lower limbs for adults for Hy. lusitanicum)
In lines 272-280, there are missing Chi-square tests values and degrees of freedom (please change it like you did in lines 253, 256-257). Also please change “pvalue” to “p”.
277-280: The statement is not true also for D. marginatus for which the ratio between men and women is even. “We observed that almost for all species, the number of ticks removed from men was significantly higher than from women, except for H. punctata and D. marginatus. There were significant differences for all species (χ2=226.675, df=7, p=0)”. - based on my calculations
Lines 281-282 and elsewhere (e.g. discussion): please keep to two digits after a dot in decimals
Author Response
Thank you very much for your throughout this process.
